# Unripe Papaya By-Product: From Food Wastes to Functional Ingredients in Pancakes

**DOI:** 10.3390/foods10030615

**Published:** 2021-03-14

**Authors:** Waralee Joymak, Sathaporn Ngamukote, Praew Chantarasinlapin, Sirichai Adisakwattana

**Affiliations:** 1Food and Nutrition Program, Department of Nutrition and Dietetics, Faculty of Allied Health Science, Chulalongkorn University, Bangkok 10330, Thailand; waralee.jm@gmail.com; 2Phytochemical and Functional Food Research Unit for Clinical Nutrition, Department of Nutrition and Dietetics, Faculty of Allied Health Science, Chulalongkorn University, Bangkok 10330, Thailand; sathaporn.n@Chula.ac.th (S.N.); praew.c@chula.ac.th (P.C.)

**Keywords:** unripe papaya, flour, fruit processing waste, functional ingredient, pancakes

## Abstract

Papaya is one of the most economic and valuable fruits in tropical countries. However, the fruit processing industries generate a high volume of unripe papaya waste and by-products. To reduce this waste, unripe papaya powder (UPP) was manufactured and incorporated into pancake formulation. The results showed that a particle size of UPP was 140.8 ± 2.1 µm, which contained polyphenolic compounds, dietary fiber and demonstrated ferric reducing antioxidant power (FRAP). Compared with wheat flour, UPP had higher values of water absorption index, water solubility index and swelling index and lower level of amylose. In the cholesterol-reducing effect, UPP decreased the formation of cholesterol micellization and bound bile acids. Interestingly, incorporation of 5–20% UPP into pancakes could decrease the glucose release with a concomitant increase in the percentage of undigestible starch. The hardness and chewiness of pancake was increased with a higher amount of UPP (10–20%). The results suggest that UPP from fruit processing waste can be regarded as a promising functional ingredient to incorporate with pancakes.

## 1. Introduction

Carbohydrates, found in foods such as fruits, vegetables, grains, and dairy products, are one of three macronutrients as a favorite source of energy. Emerging evidence suggests that long-term consumption of high carbohydrate diets has been linked to increased risks of hyperglycemia [1]. Furthermore, it is proven that long-term hyperglycemia is a main contributing factor to the development of diabetes and its complications, including blindness, kidney failure, cardiovascular diseases, stroke, and amputation [2]. With a high prevalence of diabetes, the current dietary guideline recommends low-carbohydrate consumption, which is an effective way of sustainable improvement in glycemic control [3]. Nowadays, natural food sources are widely used as a healthier food choice for glycemic control because of their enriched containing nutrients, including resistant starch, dietary fiber and phytochemical compounds. For example, partial wheat replacement with unripe banana flour and black rice flour can enhance total phenolic compounds and antioxidant capacity in bakery product [4,5,6]. Moreover, the addition of natural ingredients containing dietary fibers into food products can delay the key steps of carbohydrate and lipid digestion [7]. Pancakes, a kind of bread made on a frying pan, are one of the most popular breakfasts worldwide. However, it is produced mainly by mixing ingredients with a high content of carbohydrate and low amounts of fiber and phytochemical constituents. Therefore, replacing or mixing pancake flour with other ingredients such as fruits and vegetables is suggested to be an alternative approach to reduce starch digestibility and to increase the content of fiber and phytochemical compounds.

Papaya is widely cultivated in most tropical countries and islands. Both ripe and unripe fruit of papaya have culinary use because of its nutritional value and taste acceptability. It was found that papaya contains high levels of vitamin C, β-carotene, and dietary fiber. Interestingly, unripe papaya has lower sugar and a higher content of fiber than ripe papaya [8]. In the food industry, papaya can be consumed as fresh fruits and processed food, for example dried papaya, candy, and ice cream [9]. However, only the middle part of unripe papaya is usually chosen for preparing dried papaya products, but by-products generated during processing, including the upper and lower parts of them are discarded as a waste. To concern the environmental issue, utilization of papaya wastes becomes a major challenge and the aspects to deal with. Due to the nutritional and functional properties of unripe papaya, there has been great interest in developing alternative fiber-enriched flour and its food application. Therefore, the objective of the current study was to develop unripe papaya powder (UPP) as a by-product generated from food processing. In addition, physicochemical and functional properties of UPP were also determined. Furthermore, this research was to develop the functional pancakes with partial replacement of UPP. Finally, this product was further analyzed for in vitro starch digestibility. This would be a new application of unripe papaya flour in the food industry and would help in the development of functional foods.

## 2. Materials and Methods

### 2.1. Materials

The top and bottom part of unripe papaya waste was obtained from KCG corporation Co., LTD., Bangkok, Thailand (Figure 1A). Folin-Ciocalteu reagent, 2,2-diphenyl-1-picrylhydrazyl (DPPH), TPTZ (2,4,6-tripyridyl-s-triazine), porcine bile extract, pepsin from porcine gastric mucosa powder (250 U/mg), α-amylase Type VI-B from porcine pancreas (15.8 U/mg), pancreatin from porcine pancreas (4 × U.S. Pharmacopeia (USP) specifications), taurocholic acid, glycodeoxycholic acid, and taurodeoxycholic acid were obtained from Sigma-Aldrich Chemical Co. Ltd. (St. Louis, MO, USA). Amyloglucosidase (6 U/mg) from *Aspergillus niger* was purchased from Roche Diagnositics (CityIndianapolis, IN, USA). The glucose oxidase-peroxidase (GOPOD) and cholesterol test kit was purchased from HUMAN GmbH (Wiesbaden, Germany). A total bile acid kit was purchased from GenWay Biotech Inc. (San Diego, CA, USA).

### 2.2. Unripe Papaya Powder 

Unripe papaya powder (UPP) was prepared following a previously described method with minor modification [10,11]. Briefly, papaya was soaked with 0.7% NaCl for 30 min, then washed with water and cut into small pieces. They were placed onto aluminum foil in the tray, then heated by a hot air oven at 70 °C for 12 h. After heating, dried unripe papaya was milled by the laboratory mill for 2 min, sieved with a mesh no.100. UPP was kept in aluminum foil, placed in a zipper-lock bag and stored at −20 °C.

### 2.3. Physicochemical Properties

#### 2.3.1. Bulk Density Measurement

The samples were calculated for bulk density by mass/volume, as described in a previous report [12]. The powder was loaded into a graduated cylinder to 500 mL and weighed. The volume was read directly from the cylinder.

#### 2.3.2. Water Absorption, Solubility Index and Swelling Power

Water absorption index (WAI), water solubility index (WSI) and swelling power (SP) were done according to a previous report [13]. Briefly, the powder (50 mg) was dispersed in 1 mL of distilled water, then heated at 90 °C for 10 min. After cooling, the sample was centrifuged at 12,000 rpm at 4 °C for 5 min. The supernatant was transferred into microtube and was recovered by heating at 105 °C until constant weight. WAI, WSI and SP were calculated as follows:WAI (g/g) = (Wr)/Wi(1)
WSI (g/100g) = (Ws)/Wi × 100(2)
SP (g/g) = (Wr)/(Wi − Ws)(3)
where Ws = weight of dry solids in dried supernatant (g); Wr = weight of wet solids remaining after centrifugation (g); and Wi = weight of initial dried powder sample (g). 

#### 2.3.3. Color Measurement

The color of flour samples was determined using a CIE Hunter Lab colorimeter. The colorimeter was standardized with black glass and white calibrated tile. The powder (500 mg) was placed in sample dish above the port insert. The results were expressed in lightness (*L** value), redness (*a** value) and yellowness (*b** value).

#### 2.3.4. Microstructural Properties

The morphology and surface appearance of UPP were observed under scanning electron microscope (SEM). The sample was coated with a gold layer by an ion sputter instrument prior observation under SEM at 15 kV. The mean particle size of UPP was determined using a laser diffraction-based Malvern particle size analyzer Mastersizer 3000. A refractive index of flour and dispersant were chosen from the reference manual of Malvern instrument: 1.33 for water and 1.53 for flour. The results were expressed as a volume-weighted mean diameter. 

#### 2.3.5. Amylose Content

The determination of amylose content was done according to a previous report [14]. Briefly, the sample (10 mg) was dispersed in 100 μL of 95% ethanol and 100 μL of 1 M NaOH, then incubated at room temperature for 10 min. The sample was heated at 100 °C for 10 min and cooled for 2 h at room temperature. The sample was diluted with distilled water to a final concentration of 10 mg/mL. Thereafter, the sample (50 μL) was incubated with distilled water (910 μL), 1 N acetic acid (20 μL) and iodine solution (20 μL) for 20 min at room temperature. Finally, the absorbance was measured at 620 nm wavelength. Pure amylose from potato was used as a standard curve and expressed as a percentage of amylose content.

### 2.4. Proximate Analysis 

The proximate analysis of UPP including moisture content, ash, protein, total fat and total dietary fiber were carried out using the AOAC method (2012). Carbohydrate content was determined by calculation of the following equation: Carbohydrate (%) = 100 – % (protein + fat + moisture + ash). Available carbohydrates were calculated by the subtraction of the carbohydrate and dietary fiber.

### 2.5. Total Phenolic Content, Antioxidant Activity and Bile Acid Binding

The method of extraction was described by Maisarah et al. [15], with minor modification. The sample (5 g) was extracted with 80% methanol (50 mL), then incubated at room temperature with a 200 rpm shaker for 2 h. Thereafter, the mixture was centrifuged at 1000 rpm for 15 min. The supernatant was kept and dried with evaporator. Total phenolic content was determined using the Folin Ciocalteu method [15]. The sample (20 µL) was mixed with 150 µL of Folin Ciocalteu reagent, then incubated for 5 min. Next, 6% sodium bicarbonate solution (150 µL) was added and incubated for 90 min. The absorbance was read using a spectrophotometer at a wavelength of 760 nm. Gallic acid was used for the standard curve. The results were expressed as mg gallic acid equivalents (GAE)/100 g sample extract. 

FRAP assay was performed according to a previous study [16]. FRAP reagent was prepared freshly and warmed at 37 °C. The solution contained 0.3 M sodium acetate buffer, 10 mM TPTZ solution in 40 mM HCl and 20 mM of ferric chloride (FeCl_3_) in a ratio of 10:1:1 (*v*/*v*). Finally, the sample (20 µL) was mixed with 180 µl of FRAP reagent in a 96-well plate and incubated for 30 min. Spectrophotometer was used for analysis at a wavelength of 595 nm. Ferrous sulfate (FeSO_4_) solution was used for the standard curve. The results were expressed as millimole ferrous sulphate/100 g sample extract. 

The DPPH free radical scavenging method was performed according to a previous study with minor modifications [17]. 40 mg/mL of UPP (100 µL) was added to 100 µL of 0.2 mM DPPH solution in methanol, and then incubated for 30 min in the dark at room temperature. After incubation, the absorbance was measured at a wavelength of 515 nm. The results were expressed as mg ascorbic acid equivalent/100 g sample extract.

The bile acid (taurocholic acid, glycodeoxycholic acid, and taurodeoxycholic acid) binding assay was performed according to the previously described method [18]. Briefly, the sample (final concentration 2 mg/mL) was incubated with each bile acid including taurocholic acid, glycodeoxycholic acid and taurodeoxycholic acid (2 mM) in 0.1 M phosphate buffered saline (PBS), pH 7, at 37 °C for 90 min. After that, the mixtures were separated bound form of bile acids by filter through 0.2 µm nylon filter. The free bile acids were determined using the bile acid analysis kit (Genway biotech. Inc, San Diego, CA, USA). The absorbance was recorded at 540 nm. Sodium carboxymethyl cellulose (CMC) was used as a positive control in this study.

Artificial cholesterol micelles used as a model for in vitro cholesterol solubilization was assayed according to the method of Adisakwattana et al. [18]. Artificial micelles were imitated the natural mixed micelle by containing sodium taurocholate, egg lecithins, cholesterol, and oleic acid. First, 2 mM cholesterol, 1 mM oleic acid, and 2.4 mM phosphatidylcholine were dissolved in methanol and dried under nitrogen. The mixture was added to 15 mM PBS containing 6.6 mM taurocholate salt, at pH 7.4 and sonicated twice for 30 min. The micelle solution was incubated at 37 °C for 16 h. The sample (final concentration 10 mg/mL) was incubated with the mixed micelle at 37 °C for 2 h. The mixture was centrifuged at 10,000 rpm for 30 min, and then collected the supernatant for determination of cholesterol by total cholesterol test kit. The absorbance was recorded at 500 nm. Sodium carboxymethyl cellulose (CMC) was used as a positive control in the study.

### 2.6. Pancake Preparation

UPP (5–20%) substituted for wheat flour pancakes on an equal weight basis of the total flour in the mixture. A mixture of dry ingredients included flour (112 g), sugar (40 g), vegetable oil (21.5 g), baking powder (21.5 g), salt (1 g) and egg (56.5 g) to make a batter slurry for 15 min at room temperature. Then, the frying pan was heated with medium heat and 1 tablespoon of batter and browned on both sides for 1 min.

### 2.7. Texture Analysis of Pancake

The texture of pancake was determined by a texture analyzer (TA. XT plus texture analyzer, Stable Micro Systems, UK). Hardness, cohesiveness, springiness and chewiness were determined. A double bite compression test was performed and equipped with cylinder probe (SMPS/100). The pancake was placed on a flat base and compressed to a fixed height of 5 mm and recorded using the condition from a previous study with some modifications [19]. The condition was pretest speed: 1.0 mm/s, test speed: 1.0 mm/s, posttest speed: 1.0 m/s, compression distance: 25% and trigger force: 0.098 N.

### 2.8. Color Measurement of Pancake

The surface color of the pancakes was performed by a colorimeter. The colorimeter was standardized with black glass and white calibrated tile. After that the pancake was placed over the port insert and recorded the color of the surface pancake represented as *L**, *a** and *b** value.

### 2.9. Determination of Total Starch

The total starch content was determined following a previous report with minor modifications [20]. The pancakes were weighted 50 mg in screw-cap tube and mixed with 2 M KOH. Then, the mixture was incubated at room temperature for 1 h. After incubation, 0.4 M sodium acetate buffer was added and the pH was adjusted to 4.75. Next, amyloglucosidase (3260 U/mL) was added, incubated at 60 °C, and shaken at 100 rpm for 45 min. The mixture was aliquoted and heated at 100 °C in a heat block for 10 min, then centrifuged at 13,000 rpm for 5 min. A glucose oxidase kit determined the concentration of glucose. The glucose concentration was converted into starch by multiplying 0.9 [21]. The total starch (TS) was calculated and presented in g/100 g sample.

### 2.10. In Vitro Starch Digestion of Pancake

The procedure of in vitro digestion was performed according to a previous study with minor modifications [22]. The sample (500 mg) was mixed with 1 mL of porcine α-amylase (250 U/mL), Type VI-B. Then, the pepsin solution (4500 U/mL) was added into the mixture and incubated at 37 °C in a shaking water bath at 100 rpm for 1 h. The mixture was neutralized by adding 5 mL of 0.02 M NaOH, and 25 mL of 0.2 M sodium acetate buffer was added afterward. Finally, 5 mL of the mixture of pancreatin (2 mg/mL) and amyloglucosidase (28 U/mL) was added into the mixture and incubated for 180 min. The aliquot was kept at the difference time point for 0–180 min, then heated at 100 °C in heat block for 10 min and centrifuged at 13,000 rpm for 10 min. The aliquots were kept at −20 °C for further analysis. The concentration of glucose was determined using a glucose kit. The area under the curve (AUC) was calculated by glucose concentration curve using the trapezoidal rule whereas hydrolysis index (HI) was calculated as the AUC of the sample as a percentage of the AUC of the reference food. The glucose concentration was converted into starch by multiplying 0.9 [21]. The rapidly digestible starch (RDS): the amount of glucose released after 20 min, slowly digested starch (SDS): the amount of glucose released between 20 and 120 min of digestion and undigested starch or resistant starch: the amount of glucose over 120 min were calculated from the described equations [21,23].
RDS (%) = (G20 − FG)/TS × 100(4)
SDS (%) = (G120 − G20)/TS × 100(5)
Undigestible starch (%) = (TS − (RDS + SDS))/TS × 100(6)
FG: glucose content after 0 min of digestion
G20: glucose content after 20 min of digestion
G120: glucose content after 120 min of digestion
TS: total starch

### 2.11. Total Phenolic Content and Antioxidant Activity of Pancake

The sample extraction, total phenolic content and FRAP assay were carried out according to the abovementioned method.

### 2.12. Statistical Analysis

The results are expressed as a mean ± SEM, *n* = 3. Data was analyzed using independent sample t-tests to compare the mean between the UPP and wheat flour for bulk density, hydration property and antioxidant activity. One-way ANOVA, followed by Duncan’s post hoc test, was employed for the significant differences among the group of pancakes for texture analysis, color measurement, starch content and antioxidant activity (*p* < 0.05). All statistical analyses were conducted using SPSS version 22.0.

## 3. Results and Discussion

### 3.1. Physicochemical Properties of Unripe Papaya Powder

The photographs of the top and bottom part of unripe papaya waste, unripe papaya powder (UPP), and wheat flour are illustrated in Figure 1A–C, respectively. The scanning electron micrographs of UPP are shown in Figure 1D,E. At a magnification of 500× and 2000×, the irregular shape of UPP was observed. The mean particle size of UPP was 140.8 ± 2.1 µm with a bimodal distribution (Figure 1F). Our findings showed that UPP had a different range of particle sizes with a similar bimodal distribution when compared to other powders from waste products such as pineapple stem (9.96 µm), unripe banana (80–156 µm) and passion fruits (<400 µm) [5,24,25]. In addition, the average particle size of wheat flour was approximately 200 µm [26]. Our findings agree with the previous study, indicating that different particle diameters of plant starches are detected due to various techniques of milling processes [27].

According to the proximate composition of UPP, the major components were 77.16% carbohydrate, contributing about 56.14% of dietary fiber and 21.02% of available carbohydrates. The other major components were followed by 11.61% moisture, 5.22% ash, 4.65% protein, and total fat was the least components (1.36%). 

As shown in Table 1, UPP had higher WAI, SP and WSI than WF. However, there was no significant difference in bulk density between UPP and wheat flour. Moreover, UF had lower amylose content than WF. Hydration properties of flour are water absorption index, water solubility index and swelling power. WAI dependents on the hydrophilic group of components and the gel formation [28,29], which refers to the degree of starch gelatinization components [30]. In addition, WSI refers to the solubility of organic components, which can have an excess of water such as the presence of starch and protein content [31,32]. Our findings found that UPP had significantly higher WAI, WSI and SP than wheat flour. This could be indicative of the fact that UPP has a high capacity to hold water and contains high water-soluble components. 

The irregular shape and aggregate structure of UPP might facilitate hydration properties by increasing water absorption and swelling property [33]. The high swelling power of UPP could be attributed to their relatively high carbohydrate, dietary fiber content, size and chemical composition, which have the ability to absorb water molecules [34]. This is consistent with the results of proximate analysis, indicating that UPP contains high carbohydrate and dietary fiber. The current findings demonstrated that UPP contained higher total dietary fiber (56.14 g/100 g) than other by-product processing such as wheat and rice bran (27–45 g/100 g), peach and orange concentrate (30–37 g/100 g) [35]. The moisture content of flour was still less than 15%. This assured the quality of flour, indicating that it would not affect the storage quality or increase the proliferation of microorganism, insect infestation and agglomeration [36,37,38]. The results also demonstrated that UPP exhibited a light-yellow color with coarser particles, whereas wheat flour presented a white color with fine particles (Figure 1D). This is consistent with the results of color measurements, indicating that UPP had lower lightness and higher redness and yellowness than WF (Table 2).

### 3.2. Total Phenolic Content, Antioxidant Activity, Bile Acid Binding and Cholesterol Micellization 

Total phenolic content, the FRAP value, DPPH free radical scavenging activity, bile acid binding and inhibition of cholesterol micellization of UPP are presented in Table 2. The results exhibited that UPP had a higher content of polyphenolic compounds, the FRAP value, DPPH free radical scavenging activity than WF. It has been shown that powder made from papaya contains phytochemical components such as β-carotene, vitamin C, procyanidin, gallic acid, catechin, p-coumaric acid, epicatechin and quercetin [39,40]. These components demonstrated antioxidant activity by acting as a reductant in a redox-linked reaction, wherein Fe^3+^ is reduced to Fe^2+^ at a low pH [41]. In addition, β-carotene can quench singlet oxygen and scavenge free radicals against lipid peroxidation [42]. It is suggested that phytochemical compounds are partly responsible for the antioxidant activity of UPP [43]. 

The results demonstrated that UPP had the ability to bind both primary bile acid (taurocholic acid) and secondary bind acid (glycodeoxycholic acid and taurodeoxycholic acid). In primary bile acid, UPP (2 mg/mL) had a similar percentage of taurocholic acid binding to CMC (2 mg/mL). However, UPP had a lower percentage binding of secondary bile acids (glycodeoxycholic acid and taurodeoxycholic acid) than CMC. In addition, the percent inhibition of cholesterol micellization by UPP (10 mg/mL) was 1.5-fold higher than that of CMC (10 mg/mL). The key steps of lipid digestion and absorption involves multiple steps, including emulsification, hydrolysis and micellization [44]. The binding of bile acid and interruption of the micelle formation are important steps in the reduction of cholesterol absorption [45]. The findings showed that UPP reduced cholesterol micellization and bound bile acid. We suggest that UPP had the ability to bind bile acid, which might be related to its dietary fiber content. It has been shown that dietary fiber directly binds primary bile acid in the small intestine, resulting in increasing bile acid loss from fecal matter. This effect leads to the demand for new primary bile salt, which is synthesized by cholesterol as a precursor, and a consequent reduction of plasma cholesterol level [46].

### 3.3. Physiochemical Properties and Antioxidant Activity of Pancake with Unripe Papaya Powder

The appearance of pancakes produced from UPP is shown in Figure 2. The results showed that the addition of UPP (10% and 20%) into pancakes increased the level of total phenolic content, as presented in Table 3. Moreover, an increase in the FRAP value of pancakes with 10% and 20% UPP was markedly observed.

The textural profiles of pancake with UPP are presented in Table 4. The replacement of wheat flour with UPP affected the texture profile of pancake. The hardness and chewiness of pancakes increased with a higher ratio of UPP replacement. It could be observed that pancake with 10% UPP replacement had significantly higher values of hardness and chewiness than the control pancake. However, there were no significant differences in springiness and cohesiveness between pancake with UPP and the control pancake. The findings suggest that pancakes with a higher ratio of replacement presented an increase in the hardness value. Previous studies suggest that dietary fiber can increase water absorption and are attributed to the dilution of gluten, resulting in the increase in hard texture [47]. Moreover, the particle size and morphology are an important factor that could affect textural properties. The previous study reveals the effect of particle size on textural properties of the final product might be attributed to the contact surface through physical and chemical interactions [48]. The elevation of particle size might provide a limit contact surface, which could lead to a restriction in chemical reactions. Consequently, the larger particle size could decrease the interaction of gluten network and formation [49,50,51]. Furthermore, the effect on morphology on the textural properties was reported in relation to the disruption of granule as irregular shape could increase the water absorption, leading to the dilution of the gluten network [51]. However, the replacement of wheat flour by UPP results in a reduction in the formation of gluten network of pancakes. There is a consequent effect on hardness caused by the lack of a gluten network in food [52]. Incidentally, chewiness is represented for the force needed to disintegrate a food to swallow. This value can be calculated from hardness by multiplying it with cohesiveness and springiness [20]. Therefore, the elevation of hardness is associated with an increasing chewiness. 

The results of surface color are shown in Table 4. As expected, the addition of UPP altered the surface color of the pancake. Pancake replacement with UPP had a darker shade of brown color on the surface than the control pancake. Lightness (*L**) of pancake was decreased by increasing the UPP replacement. In contrast, redness (*a**) and yellowness (*b**) were significantly increased when the level of replacement increased. The results showed that 20% UPP replacement had the highest redness and yellowness and the lowest lightness among all the pancakes. The color of flour could be one of the important factors for the consumer’s selection [12,53]. It suggests that the light-yellow color of UPP might be partly attributed to the color of β-carotene. This is due to the yellow color of UPP incorporated with the panfrying, resulting in the darker shade of brown color in the pancake. As mentioned above, UPP had a β-carotene content corresponding to yellow-orange color, which could result in final products. Similar results observed in other studies indicate that the incorporation of flour color might be attributed to the color of the final product [12,54]. For example, chiffon cake with black rice powder showed a darker color, because of its color of black rice powder [6]. Cookies with the addition of rosehip pomace presented higher redness than control cookies [53]. Furthermore, the brown shade color on surface pancake might come from the non-enzymatic browning reaction called Maillard reaction. This reaction occurs between the carbonyl group of reducing sugar and amino acid, resulting in the production of brown polymer and melanoidins [55]. The ingredients of pancake, including sugar, milk and egg contain both protein and sugar; hence, it might contribute to the increasing of brown color by the Millard reaction.

### 3.4. In Vitro Starch Digestion of Pancake 

Four formulations of pancake were employed for in vitro starch digestion (Figure 3A). At the individual time point, pancake with UPP (5–20%) presented a slower release of glucose concentration when compared to the control, indicating that pancakes with UPP (5–20%) significantly attenuated starch digestion. As shown in Figure 3B, the replacement of wheat flour by UPP demonstrated reduction of iAUCs for glucose in pancakes. The hydrolysis index had a significant decrease in the groups with UPP replacement (Figure 3C). 

Pancakes were also determined for rapid digestible starch (RDS), slow digestible starch (SDS) and undigestible starch, and total starch (TS). The total starch content of pancake ranged between 16.49 ± 0.25 to 20.96 ± 0.52 g/100 g sample. There were no significant changes in RDS and SDS of pancakes with UPP when compared to the control pancake (Figure 3D). Interestingly, pancakes with 10% and 20% UPP significantly increased undigestible starch when compared to the control pancake. Pancakes with UPP replacement could slow down the glucose release and increased the content of undigestible starch. Dietary fiber has protective potential on the digestion and absorption of carbohydrate. The reason for explanation of these findings is that UPP contained a higher amount of fiber, which might help to reduce the starch digestibility and hydrolysis index as well as increase the proportion of undigestible starch. Brennan et al. stated that the viscosity behavior of dietary fiber could improve the glycemic control [56]. An increase in viscosity of food matrix with dietary fiber can interfere with interactions of digestive enzymes, resulting in the delay of starch digestion and the absorption of glucose [57,58].

## 4. Conclusions

UPP made from fruit processing waste is a potential source of phenolic compounds, dietary fiber and antioxidants, which inhibited the formation of cholesterol micellization and bound bile acids. It had higher hydration properties than wheat flour. Starch digestibility highlighted that the substitution of UPP significantly reduced the hydrolysis index of pancake, together with an increase in undigested starch. The replacement of wheat flour with UPP altered hardness, chewiness, and darker shade of brown color in pancake. Therefore, UPP can be regarded as a good source of natural antioxidants and nutritional properties for the development of healthy products.

## Figures and Tables

**Figure 1 foods-10-00615-f001:**
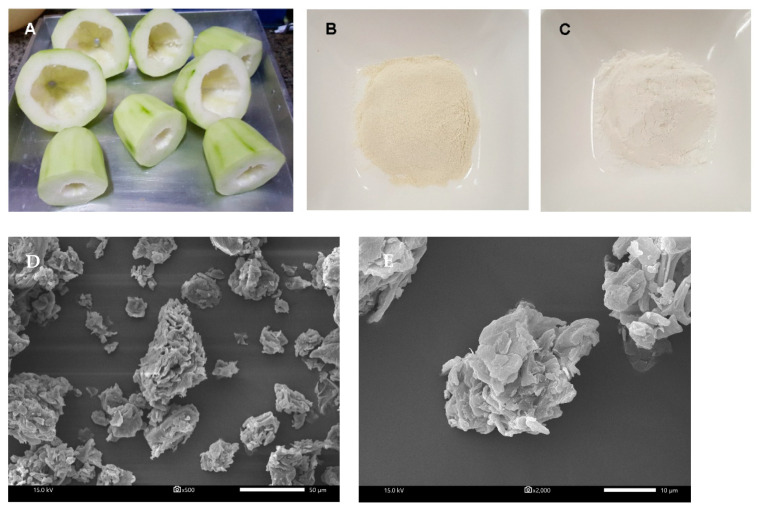
(**A**) Photographs of the top and bottom part of unripe papaya waste. The appearance of UPP (**B**) and wheat flour (**C**). Scanning electron micrograph of UPP magnified at (**D**) 500× and (**E**) 2000×. (**F**) Particle distribution by volume density of unripe papaya powder (UPP).

**Figure 2 foods-10-00615-f002:**
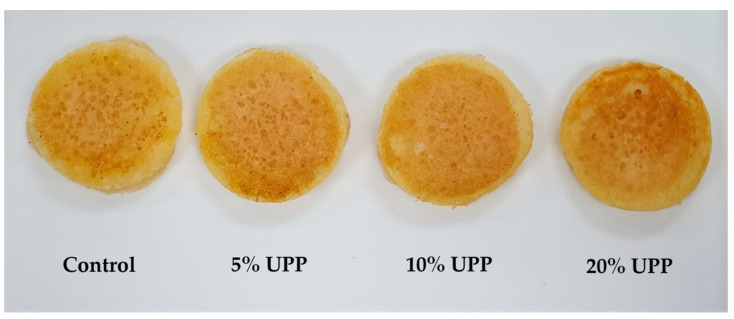
Photographs of unripe papaya pancake (5–20%).

**Figure 3 foods-10-00615-f003:**
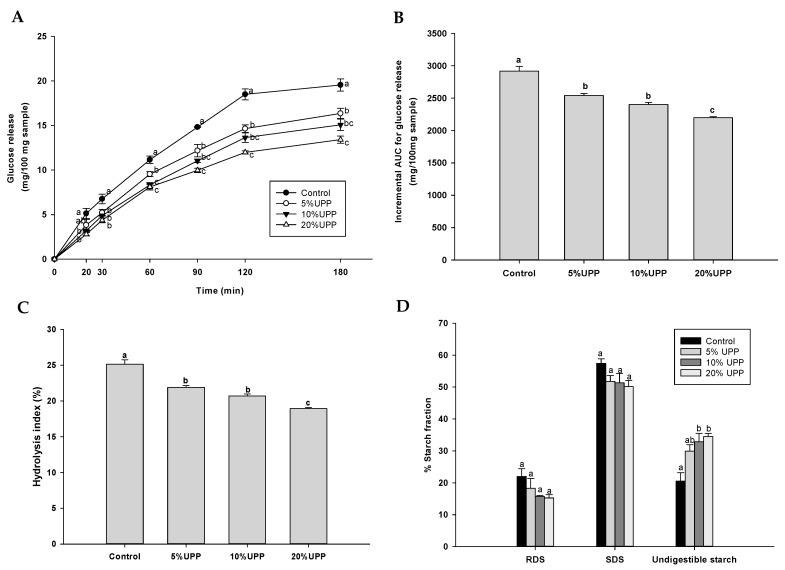
The effect of UPP replacement in pancakes on (**A**) starch digestibility, (**B**) the incremental area under the curves (iAUCs) for glucose release, (**C**) hydrolysis index and (**D**) the percentage of starch fraction including rapidly digestible starch (RDS), slowly digestible starch (SDS) and undigestible starch. Data are expressed as mean ± SEM, n = 3. Values with different letters in each column are significantly different (*p* < 0.05).

**Table 1 foods-10-00615-t001:** Hydration properties, bulk density, amylose content and color parameters of unripe papaya powder.

Samples	WAI (g/g)	WSI (g/100 g)	SP (g/g)	Bulk Density (g/mL)	Amylose Content (%)	Color Parameters		
						*L**	*a**	*b**
UPP	13.53 ± 0.11 ^a^	11.40 ± 0.18 ^a^	15.28 ± 0.1 ^a^	0.58 ± 0.01 ^a^	1.44 ± 0.33 ^a^	84.32 ± 0.01 ^a^	0.85 ± 0.00 ^a^	14.89 ± 0.01 ^a^
WF	7.51 ± 0.01 ^b^	2.93 ± 0.53 ^b^	7.74 ± 0.03 ^b^	0.53 ± 0.02 ^a^	22.66 ± 0.47 ^b^	90.11 ± 0.01 ^b^	0.42 ± 0.01 ^b^	1.02 ± 0.01 ^b^

Data are expressed as mean ± SEM, *n* = 3. Values with different letters in each column are significantly different (*p* < 0.05). WF: wheat flour; UPP: Unripe papaya power; WF: wheat flour; WAI: water absorption index; WSI: water solubility index; SP: swelling power. *L**; lightness, *a**; redness and *b**; yellowness.

**Table 2 foods-10-00615-t002:** Total phenolic content, antioxidant activity, bile acid binding and the inhibition of cholesterol micellization of unripe papaya powder.

Samples	TPC (mg GAE/100 g)	FRAP (mmol FeSO_4_/100 g)	DPPH Activity	% Bile Acid Binding			% Inhibition of Cholesterol Micellization
			(mg Ascorbic Acid/100 g)	TC	GDC	TCA	
UPP	85.67 ± 1.62 ^a^	411.58 ± 38.0 ^a^	37.87 ± 3.69 ^a^	8.35 ± 0.72 ^a^	8.46 ± 0.19 ^a^	11.89 ± 0.35 ^a^	35.0 ± 2.22 ^a^
WF	36.09 ± 1.27 ^b^	7.19 ± 0.08 ^b^	3.25 ± 0.28 ^b^	N.D.	N.D.	N.D.	N.D.
CMC	N.D.	N.D.	N.D.	8.74 ± 1.13 ^a^	18.70 ± 0.71 ^b^	21.30 ± 1.38 ^b^	22.9 ± 1.84 ^b^

Data are expressed as mean ± SEM, *n* = 3. Values with different letters in each column are significantly different (*p* < 0.05). TPC: total phenolic content; DPPH activity: DPPH free radical scavenging activity; WF: wheat flour; UPP: unripe papaya powder; TC: taurocholic acid; GDC: glycodeoxycholic acid; TCA: taurodeoxycholic acid; CMC: sodium carboxymethyl cellulose. In bile acid binding and cholesterol micellization, the concentration of UPP and CMC was 2 mg/mL and 10 mg/mL, respectively. N.D.: not determined.

**Table 3 foods-10-00615-t003:** Total phenolic content and antioxidant activity of pancakes with unripe papaya powder.

Samples	TPC (mg GAE/100 g)	FRAP (mmol FeSO_4_/100 g)
Control	14.13 ± 0.46 ^a^	31.99 ± 1.71 ^a^
5% UPP	14.58 ± 0.61 ^a^	34.50 ± 2.32 ^a,b^
10% UPP	17.37 ± 0.55 ^b^	44.27 ± 2.67 ^b,c^
20% UPP	18.65 ± 0.69 ^b^	55.07 ± 3.81 ^c^

Data are expressed as mean ± SEM, *n* = 4. Values with different letters in each column are significantly different (*p* < 0.05). Control; control pancake, 5%UPP; pancake with 5% UPP replacement, 10%UPP; pancake with 10% UPP replacement, 20%UPP; pancake with 20% UPP replacement.

**Table 4 foods-10-00615-t004:** Texture analysis and color profiles of pancakes with unripe papaya powder.

Samples	Hardness (g-Force)	Springiness (mm)	Cohesiveness	Chewiness (g-Force x mm)	Color Parameters		
					*L**	*a**	*b**
Control	220.74 ± 26.11 ^a^	0.97 ± 0.05 ^a^	0.88 ± 0.01 ^a^	192.92 ± 33.01 ^a^	68.45 ± 0.55 ^a^	7.69 ± 0.23 ^a^	45.41 ± 0.31 ^a^
5% UPP	293.63 ± 37.10 ^a^	0.92 ± 0.03 ^a^	0.86 ± 0.01 ^a^	242.05 ± 37.53 ^a,b^	66.82 ± 0.50 ^a^	9.43 ± 0.42 ^b^	42.5 ± 0.39 ^b^
10% UPP	491.46 ± 32.2 ^b,c^	0.93 ± 0.03 ^a^	0.86 ± 0.01 ^a^	398.32 ± 37.24 ^b,c^	64.13 ± 0.25 ^b^	10.63 ± 0.62 ^b^	42.11 ± 0.35 ^b^
20% UPP	568.54 ± 37.90 ^b,c^	0.93 ± 0.02 ^a^	0.85 ± 0.01 ^a^	453.36 ± 44.09 ^c,d^	62.87 ± 0.72 ^b^	14.04 ± 0.33 ^c^	40.12 ± 0.55 ^c^

Data are expressed as mean ± SEM, *n* = 4. Values with different letters in each column are significantly different (*p* < 0.05). Control; control pancake, 5%UPP; pancake with 5% UPP replacement, 10%UPP; pancake with 10% UPP replacement, 20%UPP; pancake with 20% UPP replacement. *L*;* lightness, *a**; redness and *b**; yellowness.

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
