# Peer review of "Unripe Papaya By-Product: From Food Wastes to Functional Ingredients in Pancakes"

_foods, 2021, doi:10.3390/foods10030615_

Round 1

Reviewer 1 Report

My comments are following:

  1. Material and method part should be described more in detail.
  2. Line 221: wrong style of reference.
  3. The following reference should be used since it is explaining the fortification with by product: Antonic, B., Dordevic, D., Jancikova, S., Holeckova, D., Tremlova, B., & Kulawik, P. (2021). Effect of Grape Seed Flour on the Antioxidant Profile, Textural and Sensory Properties of Waffles. Processes9(1), 131.

Author Response

Reviewer 1

My comments are following:

1.Material and method part should be described more in detail.

Re: The detail of total polyphenol, antioxidant activity, cholesterol micellization and bile acid assays were added onto the material and methods in page 4.

2.Line 221: wrong style of reference.

Re: The style was corrected.

3.The following reference should be used since it is explaining the fortification with by product: Antonic, B., Dordevic, D., Jancikova, S., Holeckova, D., Tremlova, B., & Kulawik, P. (2021). Effect of Grape Seed Flour on the Antioxidant Profile, Textural and Sensory Properties of Waffles. Processes9(1), 131.

Re: The reference was added into discussion

Reviewer 2 Report

The manuscript entitled "Unripe Papaya Powder By-Products: From Food Wastes to Functional ingredients in pancakes" reports interesting results of incorporating unripe papaya fruit powder in pancakes. It is contains well designed experiments and results are presented in good manner.

Following points should be considered during the revision of manuscript:

  1. Why authors term unripe papaya as food waste. If ripen, they will be consumed as normal foods. Food waste usually refers to unwanted/ unused parts.
  2. Line 11, please change fruit to fruits
  3. Line 48: "It found that papaya contains high vitamin C, β-carotene and dietary fiber." This sentence is not correct. Authors may have planned to write, "It was found that..."
  4. It looks like authors should change title to "Unripe papaya by-product powder:...."

Author Response

The manuscript entitled "Unripe Papaya Powder By-Products: From Food Wastes to Functional ingredients in pancakes" reports interesting results of incorporating unripe papaya fruit powder in pancakes. It is contains well designed experiments and results are presented in good manner.

Following points should be considered during the revision of manuscript:

Why authors term unripe papaya as food waste. If ripen, they will be consumed as normal foods. Food waste usually refers to unwanted/ unused parts.

Re: The reason why we refer unripe papaya as food waste was described in line 52-54. In the food manufacturing, unripe papaya is normally used only middle part, the top and bottom parts are discarded as waste.

Line 11, please change fruit to fruits

Re: It was corrected.

Line 48: "It found that papaya contains high vitamin C, β-carotene and dietary fiber." This sentence is not correct. Authors may have planned to write, "It was found that..."

Re: It was corrected.

It looks like authors should change title to "Unripe papaya by-product powder:...."

Re: It has been changed according to the comment.

Reviewer 3 Report

The present manuscript provides a characterisation of unripe papaya powder by-products technological and functional properties. This by-product has been incorporated in pancake formulation leading to a reduction in hydrolysis index of pancake together with an increase in undigested starch. Results generated provide valuable information about health-promoting properties of this substrate and may contribute to the field of by-product revalorisation. I recommend minor revision considering the points below:

-2. Materials and Methods: Have the authors considered characterizing UPP carbohydrate and phenolic fractions using more advanced techniques like chromatography?

-2. Materials and Methods, 2.4 Proximate analysis, “Carbohydrate was determined by compendium of methods for food analysis”: Please, explain. Which methods were used?

-2. Materials and Methods, 2.10 In vitro starch digestion of pancake: Was in vitro digestion method based on an official standardized protocol? Did the authors determine enzyme activity units for α-amylase, pepsin and amyloglucosidase? If so, please provide the enzyme activity assay.

-2. Materials and Methods, 2.10 In vitro starch digestion of pancake, “Then, pepsin solution (4500 U/ml; pepsin 1:3000 ex. porcine stomach mucosa, 0.8 Anson U/mg) was added into the mixture and incubated at 37°C in water bath shaking 100 rpm for 1 h”: On what basis were incubation times selected?

-2. Materials and Methods, 2.10 In vitro starch digestion of pancake, “The glucose concentration was converted into starch by multiplying 0.9”: Please, provide a reference for this factor.

-2. Materials and Methods, 2.12 Statistical analysis: since ANOVA was used to assess statistically significant differences, did all results generated follow a parametric distribution?

-3. Results and discussion, 3.1 Physicochemical properties of unripe papaya powder, “Our findings showed that UPP had the different range of particle size when compared to other powders from waste products…”: Did other powders follow a bimodal distribution?

-3. Results and discussion, 3.2 Total phenolic content, antioxidant activity, bile acid binding and cholesterol micellization: Have the authors considered comparing results from FRAP to other antioxidant activity methods like DPPH?

-3. Results and discussion, 3.2 Total phenolic content, antioxidant activity, bile acid binding and cholesterol micellization, “It has been shown that powder made from papaya contains phytochemical components such as β-carotene, vitamin C, procyanidin, gallic acid, catechin, p-coumaric acid, epicatechin and quercetin…”: Have you considered analysing these compounds in the present study?

-References: Some of the references might be outdated. In general, more references from recent articles should be added to discuss the results.

Author Response

The present manuscript provides a characterisation of unripe papaya powder by-products technological and functional properties. This by-product has been incorporated in pancake formulation leading to a reduction in hydrolysis index of pancake together with an increase in undigested starch. Results generated provide valuable information about health-promoting properties of this substrate and may contribute to the field of by-product revalorisation. I recommend minor revision considering the points below:

-2. Materials and Methods: Have the authors considered characterizing UPP carbohydrate and phenolic fractions using more advanced techniques like chromatography?

Re: In this study, authors characterized only available starch and dietary fiber which we hypothesize they act as major mechanisms to reduce starch digestibility. 

-2. Materials and Methods, 2.4 Proximate analysis, “Carbohydrate was determined by compendium of methods for food analysis”: Please, explain. Which methods were used?

Re: We removed this sentence.

-2. Materials and Methods, 2.10 In vitro starch digestion of pancake: Was in vitro digestion method based on an official standardized protocol? Did the authors determine enzyme activity units for α-amylase, pepsin and amyloglucosidase? If so, please provide the enzyme activity assay.

Re: The unit of all enzymes were provided in the materials and methods. The in vitro digestion method is based on an official standardization protocol from the previous references.

-2. Materials and Methods, 2.10 In vitro starch digestion of pancake, “Then, pepsin solution (4500 U/ml; pepsin 1:3000 ex. porcine stomach mucosa, 0.8 Anson U/mg) was added into the mixture and incubated at 37°C in water bath shaking 100 rpm for 1 h”: On what basis were incubation times selected?

Re: The incubation time at gastric period is the common protocol for 1 h. We cited the protocol on the method.

-2. Materials and Methods, 2.10 In vitro starch digestion of pancake, “The glucose concentration was converted into starch by multiplying 0.9”: Please, provide a reference for this factor.

Re: The reference was added into the end of the sentence.

-2. Materials and Methods, 2.12 Statistical analysis: since ANOVA was used to assess statistically significant differences, did all results generated follow a parametric distribution?

Re: Actually, this is the experimental design in vitro, there are commonly used at least n= 3 independent applied for One way ANOVA.

-3. Results and discussion, 3.1 Physicochemical properties of unripe papaya powder, “Our findings showed that UPP had the different range of particle size when compared to other powders from waste products…”: Did other powders follow a bimodal distribution?

Re: We added the sentence “the similar bimodal distribution to the discussion. “Our findings showed that UPP had the different range of particle size with the similar bimodal distribution when compared to other powders from waste products such as pineapple stem (9.96 µm), unripe banana (80-156 µm)”

-3. Results and discussion, 3.2 Total phenolic content, antioxidant activity, bile acid binding and cholesterol micellization: Have the authors considered comparing results from FRAP to other antioxidant activity methods like DPPH?

Re: DPPH assay was added into the materials and methods. The results of DDPH activity of UPP was added into the Table and the result section.

-3. Results and discussion, 3.2 Total phenolic content, antioxidant activity, bile acid binding and cholesterol micellization, “It has been shown that powder made from papaya contains phytochemical components such as β-carotene, vitamin C, procyanidin, gallic acid, catechin, p-coumaric acid, epicatechin and quercetin…”: Have you considered analysing these compounds in the present study?

Re: In this study, it had no idea for determination of phytochemicals due to the indicating results from antioxidant activity partly contributed to TPC.

-References: Some of the references might be outdated. In general, more references from recent articles should be added to discuss the results.

Re: Some updated references were added into the discussion.